# Epidemiology of multimorbidity in low-income countries of sub-Saharan Africa: Findings from four population cohorts

Alison J. Price[1], Modou Jobe[2], Isaac Sekitoleko[3], Amelia C. Crampin[1,4], Andrew M. Prentice[2], Janet Seeley[3,5], Edith F. Chikumbu[1], Joseph Mugisha[3], Ronald Makanga[3], Albert Dube[1], Frances S. Mair[4], Bhautesh Dinesh Jani[4]*

1 Malawi Epidemiology and Intervention Research Unit, Lilongwe, Malawi, 2 MRC Unit The Gambia @ London School of Hygiene and Tropical Medicine, Fajara, Banjul, The Gambia, 3 MRC/UVRI and LSHTM Uganda Research Unit, Entebbe, Uganda, 4 School of health and Wellbeing, College of Medicine, Veterinary and Life Sciences, University of Glasgow, Glasgow, Scotland, United Kingdom, 5 Faculty of Public Health and Policy, London School of Hygiene and Tropical Medicine, London, United Kingdom

* bhautesh.jani@glasgow.ac.uk

**Data Availability Statement:** The data that support the findings of this study are available from respective data controllers subject to successful

## Abstract

We investigated prevalence and demographic characteristics of adults living with multimorbidity (≥2 long-term conditions) in three low-income countries of sub-Saharan Africa, using secondary population-level data from four cohorts; Malawi (urban & rural), The Gambia (rural) and Uganda (rural). Information on; measured hypertension, diabetes and obesity was available in all cohorts; measured hypercholesterolaemia and HIV and self-reported asthma was available in two cohorts and clinically diagnosed epilepsy in one cohort. Analyses included calculation of age standardised multimorbidity prevalence and the cross-sectional associations of multimorbidity and demographic/lifestyle factors using regression modelling. Median participant age was 29 (Inter quartile range-IQR 22–38), 34 (IQR25-48), 32 (IQR 22–53) and 37 (IQR 26–51) in urban Malawi, rural Malawi, The Gambia, and Uganda, respectively. Age standardised multimorbidity prevalence was higher in urban and rural Malawi (22.5%;95% Confidence intervals-CI 21.6–23.4%) and 11.7%; 95%CI 11.1–12.3, respectively) than in The Gambia (2.9%; 95%CI 2.5–3.4%) and Uganda (8.2%; 95%CI 7.5–9%) cohorts. In multivariate models, females were at greater risk of multimorbidity than males in Malawi (Incidence rate ratio-IRR 1.97, 95% CI 1.79–2.16 urban and IRR 2.10; 95% CI 1.86–2.37 rural) and Uganda (IRR- 1.60, 95% CI 1.32–1.95), with no evidence of difference between the sexes in The Gambia (IRR 1.16, 95% CI 0.86–1.55). There was strong evidence of greater multimorbidity risk with increasing age in all populations (p-value <0.001). Higher educational attainment was associated with increased multimorbidity risk in Malawi (IRR 1.78; 95% CI 1.60–1.98 urban and IRR 2.37; 95% CI 1.74–3.23 rural) and Uganda (IRR 2.40, 95% CI 1.76–3.26), but not in The Gambia (IRR 1.48; 95% CI 0.56–3.87). Further research is needed to study multimorbidity epidemiology in sub-Saharan Africa with an emphasis on robust population-level data collection for a wide variety of long-term conditions and ensuring proportionate representation from men and women, and urban and rural areas.

registration and data governance application process. 1. For data access to Kiang West Longitudinal Population Study (The Gambia), please contact MRC unit The Gambia https://www.lshtm.ac.uk/research/units/mrc-gambia/research-platforms-and-clinical-cohorts 2. For data access to Malawi Epidemiology and Intervention Research rural and urban cohorts, please contact https://datacompass.lshtm.ac.uk/id/eprint/961/. 3. For data access to General Population Cohort (Uganda), please contact https://www.lshtm.ac.uk/research/centres-projects-groups/general-population-cohort.

**Funding:** This work was funded by an MRC Grant awarded to FSM (MR/T037849/1), BDJ, JS, AMP, MJ, ACC, AJP. The funder had no role in study design, data collection and analysis, decision to publish, or preparation of the manuscript.

**Competing interests:** I have read the journal's policy and the authors of this manuscript have the following competing interests: FSM – also receives funding from Wellcome, EPSRC, UKRI and CSO for multimorbidity research

## Introduction

Multimorbidity, the presence of $\geq 2$ long-term conditions (LTCs), has been identified as a current and increasing global health challenge by the United Kingdom Academy of Medical Sciences [1]. There is evidence of increasing prevalence among adults of all ages and international populations, from lowest to highest income countries [1–3] Despite growing evidence of the adverse effects of multimorbidity on health outcomes such as mortality [4], healthcare utilisation [5, 6] and quality of life [7, 8] there are substantive gaps in the literature. Most notably our lack of population-level evidence regarding multimorbidity issues in low- and middle-income countries (LMICs) [1, 2]. Gaps in the knowledge include which clusters of multimorbidity cause the greatest burden in terms of impacts on health outcomes, mortality, health care utilisation [1] and detrimental effects on quality of life. In LMICs we also have insufficient information on the determinants and temporal changes in the patterns and burden of multimorbidity. These information are crucial to inform health and social service planning across different LMIC health care systems in the context of ageing populations living with increasing levels of infectious (notably antiretroviral treated-HIV) [9] and non-infectious (e.g. hypertension, obesity, diabetes) long-term health conditions, and coexisting acute endemic infections, such as malaria, climate change impacts on health [10] and severely constrained health resources.

In LMICs, healthcare systems are likely to have additional challenges associated with the higher burden of chronic communicable diseases, which may worsen the impact of multimorbidity [11]. A scoping review on the epidemiology of multimorbidity in LMICs published in 2020 found only six studies from countries in sub-Saharan Africa out of the total of 76 studies from LMICs included in the review [12]. The current study aims to assess and harmonise available epidemiological data from three countries in sub-Saharan Africa. The objective was to identify and compare the prevalence and patterns of multimorbidity in Malawi (urban/rural), The Gambia (rural) and Uganda (rural) using available cross-sectional population-level data. Secondly, we investigated the relationship between demographic and lifestyle factors with prevalence and the patterns of multimorbidity identified and compared the key similarities and differences in the observed findings across the 3 countries and across urban vs rural communities.

## Methods

### Ethics statement

Study protocols were approved by the Malawi National Health Sciences Research Committee (*protocol # 1072*) and the London School of Hygiene and Tropical Medicine (*protocol #6303*). Study protocols were reviewed by the MRC Unit The Gambia Scientific Coordinating Committee prior to approval by the Joint Gambia Government/MRC Ethics Committee, Keneba Biobank (SCC1185v2, L2010.97v2, & L2012.31); Keneba Electronic Medical Records System (KEMReS, L2009.62); West Kiang Demographic Surveillance System (DSS, SCC961)], with the reference West Kiang Demographic Surveillance System (DSS, SCC961). Study protocols were approved by the Uganda National Council for Science and Technology (UNCST) (reference number SS998ESl) and Uganda Virus Research Institute Research Ethics Committee (reference number GC/127/828).

### Data

We used cross-sectional data from: the Malawi Epidemiology and Intervention Research rural and urban cohorts, 2013–2017 (https://datacompass.lshtm.ac.uk/961/); the Kiang West

Longitudinal Population Study (The Gambia); and the General Population Cohort in Kyamulibwa (Uganda). A brief overview of the setting, study design, participants and data sources for each study cohort is provided with greater detail elsewhere [13–15].

**Malawi.** A low-income country of Eastern Africa, approximate population 20 million people [16], with National HIV prevalence of 7.7% (95% Confidence Intervals-CI: 7.1–8.0) in those aged 15–49 years [17]. The Malawi Epidemiology and Intervention Research Unit (MEIRU) conducted a population-based prevalence survey of non-communicable conditions in rural and urban Malawi between 2013 and 2016. The rural survey was nested in the Karonga Health and Demographic Surveillance site (HDSS), a predominately subsistence economy covering 135km$^2$ in southern Karonga district, with approximately 40,000 residents. The urban survey was conducted in Lilongwe, Area 25, a high density economically mixed residential area of 23 km$^2$, with 66000 residents (25000 aged $\geq$18 years in 2008). The rural site is typical of subsistence farming and fishing communities, with comparable age and sex distributions to those observed in other districts of Malawi [14]. Area 25 is typical of other rapidly growing suburbs in the capital city of Lilongwe. All adults who were resident in one of the study areas and aged at least 18 years of age were eligible to participate. Data collection was conducted at the participant's home and prearranged to ensure that a fasted blood sample could be collected for measurement of circulating blood lipids and glucose. Written informed consent, and when appropriate assent, was obtained from all participants for interviewer-led questionnaires, anthropometry and blood pressure measurements and fasting venepuncture samples for blood glucose and lipid measurements and HIV testing. The survey instrument was modified from WHO Steps and is described in detail elsewhere [18]. In brief, the instrument was translated into local languages and data were collected using electronic tablets. Questionnaire data included age, prior diagnosis or treatment for medical conditions including hypertension, diabetes, asthma, high cholesterol, lifestyle habits including smoking and alcohol consumption and sociodemographic data including educational-level and occupation. Patient held medical records were consulted to confirm a prior diagnosis of diabetes, hypertension or high cholesterol. Protocols for measurement of blood pressure and anthropometry and venepuncture blood collection are described in detail elsewhere [14]. Participants who were identified with elevated blood pressure or fasting blood glucose (or who tested positive for HIV antibodies for the first time) were referred to the study chronic care clinic or local HIV services, as appropriate, irrespective of whether they were currently on treatment.

**The Gambia.** A low-income country of West Africa approximate population 2.9 million [19], with National HIV prevalence of 1.7% (95% CI: 1.4–2.2) in those aged 15–49 years [20]. The Kiang West Longitudinal Population Study (KWLPS) comprises >12000 individuals across 36 villages in the 750-km$^2$ rural Kiang West district of The Gambia. Available data include demographic surveillance [Kiang West Demographic Surveillance System (KWDSS)], electronic medical records [Keneba Electronic Medical Records System (KEMReS)] and biobanking platforms (Keneba Biobank), with participants' data linked through a unique West Kiang Number (WKNO) [15]. The cohort started with data collection from 3 'core' villages (approx. 4000 residents) since 1950 representing a life-course longitudinal nutritional and health phenotypes, particularly relating to anthropometry/growth and maternal health. The cohort was gradually extended to the other villages within the Kiang West district over time. Data were collected using all three databases i.e., KWDSS (from the quarterly rounds), KEMReS and the Keneba Biobank. Participants' age, sex, marital status, migration movements, and pregnancy status were derived from the KWDSS. We extracted participants' current and past medical history (diagnosis of hypertension, diabetes, asthma and epilepsy) as well as medication for those conditions from the KEMReS. The Keneba Biobank used a standardised questionnaire to collect detailed socio-demographic data including information on level of

education and assets owned. Clinical parameters (weight, height, body composition, blood pressure) were taken and fasting blood and urine samples were collected.

**Uganda.** A low-income country of Eastern Africa approximate population 45.6 million [21], with National HIV prevalence 5.2% (95% CI: 5.1–5.4) in those aged 15–49 years [22]. The MRC/UVRI and LSHTM Uganda Research Unit conducted a cross-sectional survey within a general population cohort (GPC) in Kalungu district, Kyamulibwa sub-county in rural South western Uganda between 2011–2012 [13]. The participants enrolled in the study were above 18 years of age, residing within the study area. All participants were consented before taking part in the study. Data were collected using structured standardised questionnaires on demographics and lifestyle factors i.e. sex, age, level of education, smoking status and alcohol use. Questionnaires also covered family, medical (including HIV status) and reproductive history (parity, gravidity and complications in prior pregnancies). Weight, height, waist circumference, hip circumference, and mid-upper arm circumference were measured using calibrated Seca scales, stadiometer, and flexible tape measures. After 30 minutes of rest, three blood pressure measurements, with 5 minutes' rest in between, were collected. These measurements were taken with the participant in a seated position, from the right arm when possible (otherwise collected on the left arm in those with conditions that precluded the use of the right arm), using portable sphygmomanometers (OMRON-Healthcare-Co HEM-7211-E-Model-M6; Kyoto, Japan). Non-fasting blood samples were taken from the participants in the Uganda cohort.

## Definitions

For these analyses, we defined hypertension as a systolic blood pressure measurement of at least 140mmHg, a diastolic measurement of at least 90 mmHg [23] or self-reported use of current hypertension medication or from clinical notes. Diabetes was defined as a fasting blood glucose reading of at least 7.0mmol/l [24] or prior diagnosis of diabetes (based on self-report or clinical notes), irrespective of whether the participant was currently taking anti-diabetic medication. If fasting blood glucose was not available and there was no self-reported/clinical notes confirmed diabetes diagnosis, participants were excluded from calculation of diabetes prevalence. Obesity was defined on the basis of anthropometric measurement (excluding pregnant women) as a body mass index (BMI) of at least $30.0\text{kg/m}^2$ and overweight as a BMI of $25.0$–$29.9\text{kg/m}^2$ [25]. High total cholesterol was defined on the basis of blood results as circulating cholesterol of 5mmol/L or more in a venous blood sample [26]. We defined asthma and epilepsy based on prior diagnosis or current treatment for the condition. Prior diagnosis of epilepsy (based on medical records) was available only for The Gambia data, while prior diagnosis of asthma (from self-report at interview) was available only for the Malawi data. HIV blood test status results data were available for Malawi and Uganda, and defined as test positive, test negative or unknown (if the participant had never tested or a last negative test date was more than 3 years earlier than the interview date for Malawi and more than 2 years earlier in Uganda). Pregnant women were excluded from estimates of obesity but data on health conditions including epilepsy, asthma, cholesterol, HIV, hypertension, were included in the multimorbidity estimates as these health conditions require management irrespective of pregnancy status and some conditions are either unrelated to pregnancy or may persist beyond pregnancy.

Multimorbidity was defined as the presence of two or more of the following conditions: hypertension, diabetes, HIV, obesity, asthma, epilepsy or high cholesterol. Educational attainment was categorised according to the highest level reached: none/primary, secondary or postsecondary. Tobacco smoking was defined as never, previous or current and alcohol

consumption was based on ever or never consumption within the past 12 months. These lifestyle data were not available for The Gambia.

## Statistical methods

Using the harmonised data, we summarised categorical variables using frequencies and proportions. Continuous variables were summarised using either means (with standard deviations) or medians (with interquartile ranges) based on the nature of the distribution of the variable for the different datasets i.e. Malawi (urban and rural), Uganda and The Gambia datasets. We generated a pooled and harmonised dataset to investigate associations between socio-demographic factors (age, sex and education) and lifestyle factors (smoking and alcohol consumption) and risk for multi-morbidity. We used $\chi^2$ likelihood ratio tests to assess the extent of heterogeneity in the association between risk factors and multimorbidity by cohort study to determine whether it would be necessary to analyse and present findings separately for each cohort. Age, sex, location, and education were specified *a priori* as sociodemographic risk factors of interest. We calculated age-specific prevalence of multimorbidity and the prevalence of dual combinations of the individual conditions included in the multimorbidity variable (hypertension, diabetes, asthma, high-cholesterol, HIV, obesity and epilepsy). The WHO standard population was used to generate age-standardised population prevalence estimates for comparison between the three study cohorts and with external populations.

A negative binomial regression model with a log-link function was fitted to estimate risk ratios for multimorbidity by sociodemographic and lifestyle factors, with adjustment for age and other factors including sex, education level (all three cohorts) and smoking and alcohol consumption (Malawi and Uganda only), as appropriate.). We tested departure from linearity and linear trend for age was calculated by using age as a continuous variable and by assigning each 10-year age category a dummy measure (1 to 6). We used likelihood ratio test statistic to determine the statistical significance of associations with multimorbidity. We accounted for potential clustering (because recruitment included all household adults in the study populations in each study population, and family members share factors such as socioeconomic status and diet) by calculating robust standard errors. Missing values represented less than 5% of the data in all variables and did not contribute to regression analyses. We excluded pregnant women from anthropometric analyses. All statistical analyses were performed by using STATA 15 statistical software (StataCorp LP, College Station, TX), and 2-tailed tests of statistical significance with P values <0.05 were considered significant.

## Results

A total of 44,359 individuals aged 18 years and older, resident in urban (N = 16,671) and rural (n = 13,903) Malawi (total N = 30,574), rural The Gambia (n = 7,917) and rural Uganda (n = 5,868) contributed data to these analyses, with more female than male participants in each study. Median age of participants was 29 (Inter quartile range IQR 22–38), 34 (IQR25-48), 32 (IQR 22–53) and 37 (IQR 26–51) in urban Malawi, rural Malawi, The Gambia, and Uganda, respectively. The majority of participants were aged <40 years, in The Gambia (~60%), Malawi (~60% rural, ~80% urban) and Uganda (~55%) and a greater proportion of participants were aged ≥60 years in The Gambia (~18%) and Uganda (~15%) and rural Malawi (~12%) compared to urban Malawi (~5% urban). In rural Malawi, The Gambia and rural Uganda, the majority of participants reported having completed no further than primary education (65.6, 66.4, and 72.1%, respectively), with few attaining post-secondary education (1.82, 1.78 and 5.54%, respectively). In urban Malawi, half of the participants reported completing secondary education (52.2%) and a further 16.5%% reported completion of a post-secondary

**Table 1. Baseline socio-demographic factor prevalence estimates: Malawi, The Gambia and Uganda.**

| Data | Malawi: (Urban) | Malawi: (Rural) | The Gambia (Rural) | Uganda (Rural) |
|---|---|---|---|---|
| Sample Size | 16,671 | 13,903 | 7,917 | 5,868 |
| Female participants (%) | 10,867 (65.1) | 8,039 (57.8) | 4,796 (60.5) | 3,443 (58.6) |
| Mean age (SD) | 32.4 (12.9) | 37.9 (16.4) | 38.7 (19.3) | 40.2 (16.9) |
| Median Age (IQR) | 29 (22–38) | 34 (25–48) | 32 (22–53) | 37 (26–51) |
| **Age categories (%)** | | | | |
| 18–29 | 8,539 (51.2) | 5,254 (37.7) | 3,689 (46.6) | 1,912 (32.5) |
| 30–39 | 4,444 (26.6) | 3,405 (24.4) | 1,000 (12.6) | 1,311 (22.3) |
| 40–49 | 1,835 (11.0) | 2,137(15.3) | 981 (12.4) | 1,047 (17.8) |
| 50–59 | 991 (5.9) | 1,411 (10.1) | 832 (10.5) | 711 (12.1) |
| 60–69 | 559 (3.3) | 838 (6.0) | 665 (8.4) | 446 (7.6) |
| >70 | 303 (1.8) | 858 (6.1) | 750 (9.5) | 441 (7.5) |
| **Education status (%)** | | | | |
| Pre-Primary or none | 584 (3.5) | 595 (4.3) | 2,999 (37.9) | 711 (12.1) |
| Primary | 4,632 (27.8) | 8,524 (61.3) | 2,257 (28.5) | 3,521 (60.0) |
| Secondary | 8,701 (52.2) | 4,531 (32.6) | 2,248 (28.4) | 1,308 (22.3) |
| Tertiary | 2,754 (16.5) | 253 (1.8) | 141 (1.8) | 325 (5.5) |
| Missing | 0 | 0 | 272 (3.4) | 3 (0.05) |

SD = Standard Deviation; IQR = Interquartile range.

qualification (Table 1). Table 2 shows the lifestyle factor information on tobacco and alcohol consumption, which were available for the Malawi and Uganda populations only. The majority of urban and rural Malawian (93.9 and 92.3%) and rural Ugandan participants (85.8%) were never smokers, with 3.2, 5.8 and 10.8% current smokers, respectively. In Malawi 17% of urban and 19.8% of rural participants reported alcohol consumption. In rural Uganda, 35.5% of participants reported alcohol consumption.

WHO age standardized prevalence estimates for long-term conditions are shown in Table 3. Following age standardization, prevalence of hypertension was comparable in the four study populations and the prevalence of HIV was comparable between Malawi and Uganda cohorts, with no data available for The Gambia. In contrast, the age standardised burden of diabetes and obesity was greater in urban Malawi than the other three cohorts. Similarly, while data were available only from two countries, the burden of asthma was greater in urban and rural Malawi than in The Gambia and the burden of high cholesterol was greater in urban and

**Table 2. Baseline lifestyle factor prevalence estimates: Malawi and Uganda.**

| Data | Malawi: (Urban) | Malawi: (Rural) | Uganda (Rural) |
|---|---|---|---|
| Sample Size | 16,671 | 13,903 | 5,868 |
| **Tobacco smoking status (%)** | | | |
| Never | 15,653 (93.9) | 12,826 (92.2) | 5,036 (85.8) |
| Former | 485 (2.9%) | 272 (2.0) | 194 (3.3) |
| Current | 533 (3.2%) | 805 (5.8) | 635 (10.8) |
| Missing | 0 | 0 | 3 (0.05) |
| **Alcohol consumed within last 12 months (%)** | | | |
| No | 13,836 (83.0) | 11,154 (80.2) | 3,780 (64.4) |
| Yes | 2,835 (17.0%) | 2,749 (19.8) | 2,085 (35.5) |
| Missing | 0 | 0 | 3 (0.05) |

**Table 3. WHO age-standardized prevalence estimates for single long-term conditions: Malawi, The Gambia and Uganda.**

| LTC | | Malawi (urban) | Malawi (rural) | The Gambia (rural) | Uganda (rural) |
|---|---|---|---|---|---|
| Hypertension | Number of cases (crude prevalence %) | 2,403 (14.4) | 1,884 (13.5) | 1,423 (17.9) | 1,170 (19.4) |
| | WHO age standardized prevalence % (95% CI) | 28.2 (27.3–29.1) | 17.9 (17.2–18.5) | 20.8 (19.9–21.8) | 22.0 (20.9–23.0) |
| Diabetes | Number of cases (crude prevalence %) | 401 (2.4) | 208 (1.5) | 81 (1.0) | 112 (1.0) |
| | WHO age standardized prevalence % (95% CI) | 4.9 (4.4–5.4) | 1.9 (1.6–2.2) | 1.2 (0.9–1.4) | 2.0 (1.6–2.4) |
| Asthma | Number of cases (crude prevalence %) | 864 (5.1) | 623 (4.4) | 177 (2.2) | No data available |
| | WHO age standardized prevalence % (95% CI) | 5.3 (4.9–5.8) | 4.4 (4.0–4.7) | 2.5 (2.1–2.9) | |
| High Cholesterol defined as >5 mmol/l | Number of cases (crude prevalence %) | 1,839 (11.0) | 1,707 (12.2) | No data available | 560 (9.5) |
| | WHO age standardized prevalence % (95% CI) | 18.3 (17.5–19.2) | 14.8 (14.1–15.4) | | 10.4 (9.6–11.2) |
| HIV test status | Number of cases (crude prevalence %) | 1,313 (7.8) | 1,217 (8.7) | No data available | 575 (9.8) |
| | WHO age standardized prevalence % (95% CI) | 10.1 (9.5–10.7) | 9.8 (9.3–10.3) | | 9.9 (9.1–10.7) |
| Obesity[1] defined as BMI >30kg/m$^2$ | Number of cases (crude prevalence %) | 2,070 (12.4) | 631 (4.5) | 146 (1.8) | 195 (3.3) |
| | WHO age standardized prevalence % (95% CI) | 17.1 (16.3–17.9) | 5.4 (5.0–5.8) | 2.5 (2.1–2.9) | 3.4 (3.0–3.9) |
| Epilepsy | Number of cases (crude prevalence %) | No data available | No data available | 53 (0.67) | No data available |
| | WHO age standardized prevalence % (95% CI) | | | 0.66 (0.46–0.86) | |

CI = Confidence Intervals; BMI = Body Mass Index.

[1] Excluding pregnant women N = 241

rural Malawi compared to Uganda. Epilepsy data were only available for The Gambian population. Table 4 shows the age-standardized prevalence for multimorbidity and dual combinations of long-term conditions, with evidence of greater burden of multimorbidity in urban Malawi (22.4%; 95% confidence intervals-CI 21.5–23.3%) and rural Malawi (11.7%; 95% CI 11.12–12.31) than observed in The Gambia (2.9; 2.5–3.3%) or Uganda (8.2; 95%CI 7.4–8.9%). Fig 1 shows the age and sex specific multimorbidity estimates by study population. The highest burden was observed in older men and women in all populations, however urban Malawian women had greater multimorbidity prevalence than women and men in all other populations within a comparable age group. Of note, N = 41 women were pregnant at the time of cohort recruitment and excluded from the anthropometric measurements for obesity.

There was evidence of heterogeneity in the association between sociodemographic and lifestyle factors and risk for multimorbidity by study cohort, hence results are presented separately for each cohort in Table 5. In multivariate models, females were at greater risk of multimorbidity than males in urban Malawi (Incidence rate ratio-IRR 1.97, 95% CI 1.79–2.16), rural Malawi (IRR2.1 (1.86–2.37) and Uganda (IRR- 1.60, 95% CI 1.32–1.95), with no evidence of a difference between the sexes in The Gambia. There was strong evidence of greater risk of multimorbidity with increasing age, measured continuously or by 10-year age group in all four study populations (p <0.0001 for trend). Higher educational attainment was also associated with increased risk of multimorbidity in Malawi (IRR 1.78, 95%CI 1.60–1.98 urban and 2.37, 95%CI 1.74–3.23 in rural) and Uganda (IRR 2.40, 95% CI 1.76–2.36), but not in The Gambia (IRR 1.48; 95% CI 0.56–3.87), following adjustment for age and sex. In Malawi, compared to never smokers, current smokers in urban and rural populations had lower risk for multimorbidity. Current alcohol consumers in the urban population had increased risk of

**Table 4. WHO age-standardized and age-specific multimorbidity and dual long-term conditions combinations prevalence estimates: Malawi, The Gambia and Uganda.**

| | | Malawi (urban) N = 16,671 | Malawi (rural) N = 13,903 | The Gambia (rural) N = 7,917 | Uganda (rural) N = 5,868 |
|---|---|---|---|---|---|
| Multimorbidity (any combination of 2 or more LTC) | Number of cases | 2004 | 1244 | 192 | 433 |
| | WHO age standardized prevalence: % (with 95% CI) | 22.4 (21.5–23.3) | 11.7 (11.1–12.3) | 2.9 (2.5–3.5) | 8.2 (7.4–8.9) |
| | **LTC Combinations[1]** | | | | |
| Hypertension and Diabetes | Number of cases | 233 | 107 | 55 | 49 |
| | WHO age standardized prevalence: % (with 95% CI) | 3.4 (3.0–3.9) | 1.1 (0.89–1.3) | 0.79 (0.58–1.0) | 0.92 (0.06–1.1) |
| Hypertension and Obesity | Number of cases | 668 | 242 | 65 | 62 |
| | WHO age standardized prevalence: % (with 95% CI) | 8.2 (7.6–8.9) | 2.46 (2.1–2.7) | 1.1 (0.87–1.4) | 1.1 (0.88–1.4) |
| Diabetes and Obesity | Number of cases | 159 | 47 | 8 | 8 |
| | WHO age standardized prevalence: % (with 95% CI) | 2.1 (1.77–2.49) | 0.45 (0.32–0.58) | 0.12 (0.03–0.21) | 0.15 (0.04–0.25) |
| | **LTC Combinations[2]** | | | | |
| Diabetes and High Cholesterol | Crude number of cases | 164 | 65 | Not available | 16 |
| | WHO age standardized prevalence: % (with 95% CI) | 2.3 (1.9–2.7) | 0.65 (0.49–0.81) | | 0.33 (0.16–0.49) |
| Diabetes and HIV | Crude number of cases | 50 | 20 | Not available | 7 |
| | WHO age standardized prevalence: % (with 95% CI) | 0.59 (0.42–0.77) | 0.19 (0.11–0.27) | | 0.13 (0.03–0.24) |
| High Cholesterol and HIV | Crude number of cases | | | Not available | 54 |
| | WHO age standardized prevalence: % (with 95% CI) | 2.4 (2.1–2.7) | 1.7 (1.4–1.9) | | 1.0 (0.73–1.2) |
| High Cholesterol and Obesity | Crude number of cases | 490 | 184 | Not available | 49 |
| | WHO age standardized prevalence % (with 95% CI) | 5.5 (5.0–6.1) | 1.7 (1.5–2.0) | | 0.92 (0.66–1.1) |
| HIV and Obesity | Crude number of cases | 158 | 40 | Not available | 20 |
| | WHO age standardized prevalence % (with 95% CI) | 1.4 (1.1–1.6) | 0.36 (0.24–0.47) | | 0.36 (0.20–0.51) |
| Hypertension and High Cholesterol | Crude number of cases | 635 | 497 | Not available | 200 |
| | WHO age standardized prevalence % (with 95% CI) | 9.3 (8.6–9.9) | 5.0 (4.5–5.4) | | 3.9 (3.3–4.4) |
| Hypertension and HIV | Crude number of cases | 212 | 128 | Not available | 88 |
| | WHO age standardized prevalence % (with 95% CI) | 2.3 (2.0–2.7) | 1.2 (1.0–1.4) | | 1.6 (1.3–1.9) |
| Hypertension and Asthma | Crude number of cases | 132 | 85 | 64 | Not available |
| | WHO age standardized prevalence % (with 95% CI) | 1.5 (1.2–1.8) | 0.80 (0.62–0.97) | 0.89 (0.66–1.1) | |
| Diabetes and Asthma | Crude number of cases | 26 | 8 | 5 | Not available |
| | WHO age standardized prevalence % (with 95% CI) | 0.32 (0.19–0.45) | 0.75 (0.21–0.13) | 0.08 (0.01–0.15) | |
| Obesity and Asthma | Crude number of cases | 166 | 37 | 5 | Not available |
| | WHO age standardized prevalence % (with 95% CI) | 0.4 (0.5–0.6) | 0.32 (0.22–0.43) | 0.08 (0.01–0.15) | |

[1] data for the multimorbidity variable were available from all three cohorts.

[2] data for the multimorbidity variable were available from Malawi and Uganda only

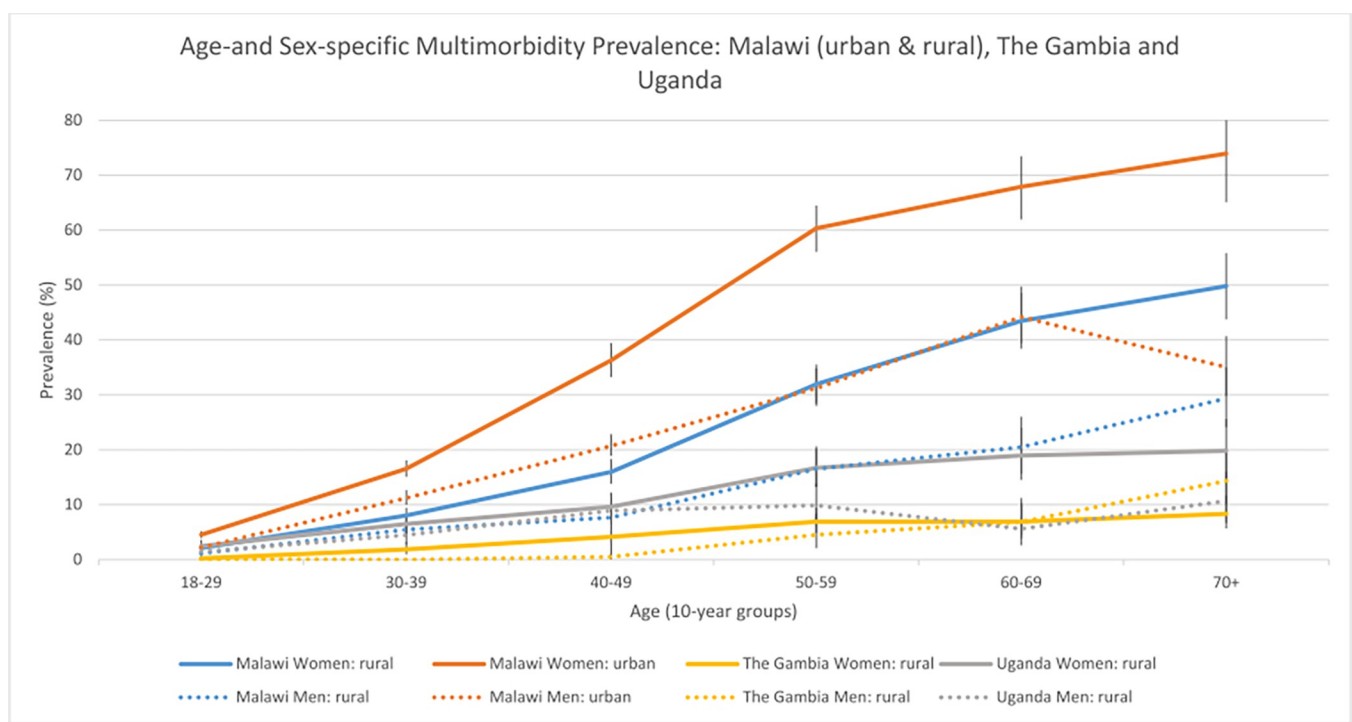

**Fig 1. Age- and sex-specific multimorbidity prevalence: Malawi, The Gambi and Uganda.**

multimorbidity compared to never consumers but this was not observed in the rural population. There was no evidence for these associations in the Ugandan data and data were not available for The Gambia on tobacco or alcohol consumption.

## Discussion

This is the first three country comparative study of multimorbidity prevalence in rural and urban sub-Saharan Africa. We observed differences in age-standardized multimorbidity burden between urban (22.5%) and rural (11.7%) Malawi, The Gambia (2.9%) and Uganda (8.2%), with the greatest burden observed in urban Malawi. Notably, hypertension comorbid with obesity was one of the most frequent combinations of LTCs. Furthermore, females were at higher risk of multimorbidity compared to males in all study populations, apart from The Gambia cohort, as observed in previous studies from high- and middle-income countries [27–29]. We observed greater multimorbidity prevalence in older age groups in each study population, consistent with previous finding from high-, middle- and low-income countries [30, 31], however, the high burden of multimorbidity observed for urban dwelling women in Malawi between 30–50 years differed from the burden observed in men and rural dwelling women. In Malawi and Uganda, those with the highest education level experienced increased risk for multimorbidity, while the least educated had a reduced risk for multimorbidity in urban Malawi and rural Uganda. Notably, the pattern of highest risk in the most educated in three of the study populations is inverse to the pattern commonly observed in high income countries, where the highest burden is among the least educated and most deprived [32]. While the drivers of these differences cannot be determined from the available data, our findings highlight likely variation in lifestyle risk factors, including differences in work-related physical activity, with the least educated more likely to engage in greater levels of physical labour (farming and

**Table 5. Estimates of the association between sociodemographic and lifestyle factors and multimorbidity: Malawi, The Gambia and Uganda.**

| | | Urban Malawi n = 12,726 | Rural Malawi n = 11,368 | Rural Gambia N = 7917. n = 6643. | Rural Uganda n = 5868 |
|---|---|---|---|---|---|
| **Model 1**[2] | | Estimates* with 95% Confidence Intervals | | | |
| **Sex: Male (reference)** | Female | 1.73 (1.58–1.90) | 1.84 (1.65–2.06) | 1.10 (0.83–1.47) | 1.60 (1.32–1.95) |
| | P value[1] | <0.0001 | <0.0001 | 0.71 | <0.0001 |
| **Age categories: Age 18–29 (reference)** | 30–39 | 3.91 (3.37–4.54) | 4.03 (3.10–5.24) | 7.43 (2.79–19.75) | 2.74 (1.86–4.03) |
| | 40–49 | 8.27 (7.15–9.56) | 7.23 (5.60–9.34) | 16.91 (7.02–40.76) | 4.56 (3.16–6.59) |
| | 50–59 | 13.40 (11.64–15.43) | 14.74 (11.53–18.85) | 31.97 (13.64–74.95) | 6.75 (4.69–9.71) |
| | 60–69 | 16.17 (13.96–18.72) | 19.57 (15.25–25.11) | 35.55 (15.07–83.85) | 6.66 (4.52–9.82) |
| | >70 | 15.79 (13.35–18.67) | 24.12 (18.83–30.89) | 55.59 (24.15–127.99) | 7.76 (5.28–11.40) |
| | P value[1] | <0.0001 | <0.0001 | <0.0001 | <0.0001 |
| **Model 2**[3] | | Estimates* with 95% Confidence Intervals | | | |
| **Sex: Male (reference)** | Female | 1.97 (1.79–2.16) | 2.10 (1.86–2.37) | 1.16 (0.86–1.55) | 1.60 (1.32–1.95) |
| | P value[1] | <0.0001 | <0.0001 | 0.31 | <0.0001 |
| **Age categories: Age 18–29 (reference)** | 30–39 | 3.97 (3.42–4.60) | 4.24 (3.25–5.51) | 8.02 (2.89–22.24) | 3.04 (2.04–4.50) |
| | 40–49 | 8.64 (7.46–10.0) | 7.92 (6.10–10.26) | 19.22 (7.29–50.65) | 5.51 (3.75–8.10) |
| | 50–59 | 14.80 (12.80–17.08) | 16.91 (13.12–21.79) | 37.59 (14.00–100.94) | 8.42 (5.74–12.36) |
| | 60–69 | 18.15 (15.61–21.09) | 22.71 (17.53–29.41) | 41.94 (14.90–118.03) | 9.32 (6.16–14.11) |
| | >70 | 19.62 (16.43–23.42) | 28.82 (22.21–37.39) | 68.82 (24.82–190.81) | 11.90 (7.80–18.17) |
| | P value[1] | <0.0001 | <0.0001 | <0.0001 | <0.0001 |
| | **Education status** | | | | |
| **Education status: Primary (reference)** | Pre-primary or none | 0.76 (0.65–0.89) | 0.93 (0.78–1.11) | 0.99 (0.60–1.63) | 0.60 (0.45–0.80) |
| | Secondary | 1.22 (1.11–1.34) | 1.48 (1.30–1.69) | 1.52 (0.79–2.94) | 1.60 (1.24–2.05) |
| | Higher than secondary | 1.78 (1.60–1.98) | 2.37 (1.74–3.23) | 1.48 (0.56–3.87) | 2.40 (1.76–3.26) |
| | P value[1] | <0.0001 | <0.0001 | 0.54 | <0.0001 |
| **Model 3**[4] | | Estimates* with 95% Confidence Intervals | | | |
| **Sex: Male (reference)** | Female | 2.18 (1.97–2.43) | 1.98 (1.72–2.27) | | 1.84 (1.47–2.30) |
| | P value[1] | <0.0001 | <0.0001 | | <0.0001 |
| **Age categories: Age 18–29 (reference)** | 30–39 | 3.96 (3.42–4.59) | 4.31 (3.31–5.62) | | 2.96 (1.99–4.41) |
| | 40–49 | 8.74 (7.55–10.12) | 8.02 (6.18–10.41) | | 5.35 (3.62–7.91) |
| | 50–59 | 15.25 (13.21–17.62) | 17.21 (13.35–22.19) | | 8.16 (5.52–12.08) |
| | 60–69 | 18.98 (16.33–22.05) | 22.91 (17.68–29.65) | | 9.05 (5.93–13.81) |
| | >70 | 20.78 (17.39–24.80) | 28.78 (22.19–37.33) | | 11.72 (7.63–17.99) |
| | P value[1] | <0.0001 | <0.0001 | | <0.0001 |
| **Highest of education: Primary (reference)** | Pre-primary or none | 0.75 (0.64–0.88) | 0.94 (0.79–1.12) | | 0.60 (0.44–0.80) |
| | Secondary | 1.21 (1.10–1.33) | 1.47 (1.29–1.68) | | 1.61 (1.25–2.07) |
| | Higher than secondary | 1.73 (1.55–1.93) | 2.31 (1.70–3.14) | | 2.40 (1.77–3.27) |
| | P value[1] | <0.0001 | <0.0001 | | <0.0001 |
| **Smoking status: Never (reference)** | Former | 1.07 (0.82–1.38) | 1.17 (0.86–1.58) | | 0.94 (0.56–1.57) |
| | Current | 0.58 (0.41–0.82) | 0.45 (0.30–0.65) | | 1.05 (0.76–1.46) |
| | P value[1] | 0.016 | 0.0001 | | 0.93 |
| **Alcohol intake in past 12 months: No (reference)** | Yes | 1.59 (1.40–1.80) | 1.06 (0.89–1.26) | | 1.12 (0.93–1.35) |
| | P value[1] | <0.001 | 0.53 | | 0.28 |

*Estimates are Incidence rate ratios

[1] *P* values were calculated using likely hood ratio test statistics and represents a test for trend for age, using age as a continuous variable. p

[2] Model 1: negative binomial models were adjusted for age (18, 29, 30–39, 40–49, 50–59, 60–69, 70+) or sex (male, female), as appropriate.

[3] Model 2: negative binomial models were adjusted for age (18, 29, 30–39, 40–49, 50–59, 60–69, 70+), sex (male, female) and education level (none or pre-primary, primary, secondary, tertiary), as appropriate

[4] Model 3: negative binomial models were adjusted for age (18, 29, 30–39, 40–49, 50–59, 60–69, 70+), sex (male, female), education level (none or pre-primary, primary, secondary, tertiary), alcohol consumption in past 12-months (no, yes) and tobacco consumption (never, former current), as appropriate

fishing lifestyles) and to have reduced access to processed, energy dense foods. Similar findings have been observed in a observational study conducted in north rural India [33] however, a systematic review of 19 studies from Southeast Asia found inconsistent associations between education level and multimorbidity prevalence [34].

The variation in prevalence estimates observed between our four study populations is consistent with previous studies using data from multiple countries, including those from Africa [30, 31, 35]. For example, the WHO study on global ageing and adult health (SAGE) reported a pooled multimorbidity prevalence of 23% using data from 44,715 community dwelling adults living in China, India, Mexico, Russia, South Africa, and Ghana [36]. However, the burden varied substantively between settings: South Africa (22.8% urban and 13.7% rural) and Ghana (24.8% urban and 12.4% rural) [36]. A meta-analysis of 193 studies, including one study from Ethiopia [37] and three studies from South Africa [38–40] and one from Ethiopia [37] reported a pooled African multimorbidity (crude) prevalence of 13.8% (4.5–35.2). However individual study prevalence estimates ranged from 2.7% (in the community) [40] to 35% (in primary care) [38]. Other individual multimorbidity studies conducted in sub-Saharan Africa have observed differences in multimorbidity prevalence [12], some of which might be explained by variation in the research methodology and study setting. Studies in Ghana [41], Nigeria [42], and Ethiopia [37] have been conducted in secondary care, using self-reported [37, 41] or clinical records data [42] to determine crude prevalence estimates. Similarly, a study in Zimbabwe in HIV patients in secondary care reported using combined self-report and measured data to determine crude prevalence estimates [43]. In contrast, studies in Burkina Faso [44] and Malawi [14] have been conducted in the community and although these studies reported similar crude multimorbidity prevalence (of 65%), there were methodological differences, including eligibility criteria (adults aged >60 years in Burkina Faso and > = 18 years in Malawi) and data sources (self-report only in Burkina Faso and a combination of self-report and medical notes in Malawi). In a small study in rural Tanzania, self-reported multimorbidity prevalence was 26.1% (adjusting for frailty weighting) in older adults aged >60 years [45]. Notably, frailty adjusted multimorbidity prevalence in the Tanzanian study increased to 67.3% when clinical assessment/screening was undertaken, highlighting the potential for under-estimation of burden when relying on self-report [45]. Finally, in a large sample of >40-year-old community dwelling adults from rural South Africa, a much higher multimorbidity prevalence of 69.4% was observed; however, the study had information on prevalence of a much broader range of LTCs [46].

## Strengths and limitations

We described and compared, with appropriate age-standardization, the epidemiology of multimorbidity in sub-Saharan Africa in three different countries including data from four populations (one urban and three rural) and representation from a wide spectrum of age groups, men and women and including detailed self-reported and measured multimorbidity outcomes. However, there are several limitations associated with this study. Firstly, these cross-sectional data are not suitable for ascertaining causality between the predictors considered and multimorbidity prevalence due to the likely effect of reverse causation in the observed associations. There is also the risk of residual confounding due to unmeasured factors and non-differential measurement error. Importantly, information was not available on many health conditions that may be important in these study populations, including mental health, musculoskeletal and painful conditions. Furthermore, information was available on a maximum of seven conditions across the datasets so multimorbidity prevalence observed in our study is likely to be an under representation. There were some differences in the underlying long-term

conditions assessed across different countries which is likely to have impacted the overall prevalence of multimorbidity in each of the cohort. This introduces a possible source of bias when multimorbidity prevalence is controlled across different cohorts. There were some differences in protocols for measurement of anthropometric and biomarker measurements, which in turn could impact directionality and prevalence estimates (and observed variations) between study populations. Lifestyle related information was very limited and restricted to smoking status and alcohol consumption, which was available only in three study populations which limited risk factor comparisons. Detailed data on quantity, duration and pattern of tobacco and alcohol consumption are needed to understand better the observed associations, notably the observed protective effect of current smoking. Importantly, information on other lifestyle factors (such as dietary intake, air pollution and physical activity levels) were missing, which are likely to have significant impact on multimorbidity prevalence [47]. Previous evidence suggests that effects of urbanisation on NCD prevalence is increasing rapidly in southern sub-Saharan Africa, we were unable to explore such regional differences with the limited data available [48]. Finally, some of our health data was based on self-report rather than healthcare records which may lead to under estimations of prevalence.

## Conclusion

The burden of multimorbidity varied within and between the three sub-Saharan Africa countries. Notably, urban dwellers, females, older people (>50 years) and those with higher educational attainment experienced the greatest multimorbidity burden. Nonetheless, there was a substantive burden of multimorbidity observed in adults aged 30–50, particularly in urban dwelling women, which portends a future high burden on health services. Further research is needed to understand better the epidemiology of multimorbidity in low-income countries of sub-Saharan Africa. This includes robust data collection involving screening and clinical assessment for a wider variety of long-term conditions (including disability, other chronic communicable diseases, pain, and mental health disorders) from community populations, ensuring proportionate representation from men and women living in urban and rural areas. Such information is crucial to inform healthcare policy and delivery in sub-Saharan Africa.

## Supporting information

**S1 Checklist. For reporting cohort can be found in the supporting information file.**
(DOCX)

## Acknowledgments

The authors would like to thank the participants for their time. The authors would like to acknowledge members of the wider MAfricaEE project team: Dr. Gertrude Chapotera, Dr Christopher Bunn.

## Author Contributions

**Conceptualization:** Alison J. Price, Modou Jobe, Amelia C. Crampin, Andrew M. Prentice, Janet Seeley, Frances S. Mair, Bhautesh Dinesh Jani.

**Data curation:** Alison J. Price, Modou Jobe, Isaac Sekitoleko, Amelia C. Crampin, Andrew M. Prentice, Joseph Mugisha, Ronald Makanga, Albert Dube.

**Formal analysis:** Alison J. Price.

**Funding acquisition:** Alison J. Price, Modou Jobe, Amelia C. Crampin, Andrew M. Prentice, Janet Seeley, Frances S. Mair, Bhautesh Dinesh Jani.

**Investigation:** Alison J. Price, Modou Jobe, Isaac Sekitoleko, Amelia C. Crampin, Andrew M. Prentice, Janet Seeley, Frances S. Mair, Bhautesh Dinesh Jani.

**Methodology:** Alison J. Price, Modou Jobe, Isaac Sekitoleko, Amelia C. Crampin, Andrew M. Prentice, Janet Seeley, Frances S. Mair, Bhautesh Dinesh Jani.

**Project administration:** Frances S. Mair, Bhautesh Dinesh Jani.

**Resources:** Alison J. Price, Modou Jobe, Amelia C. Crampin, Andrew M. Prentice, Janet Seeley, Frances S. Mair, Bhautesh Dinesh Jani.

**Software:** Bhautesh Dinesh Jani.

**Supervision:** Alison J. Price, Modou Jobe, Amelia C. Crampin, Andrew M. Prentice, Janet Seeley, Frances S. Mair, Bhautesh Dinesh Jani.

**Validation:** Alison J. Price, Modou Jobe, Isaac Sekitoleko, Amelia C. Crampin, Andrew M. Prentice, Janet Seeley, Frances S. Mair, Bhautesh Dinesh Jani.

**Visualization:** Alison J. Price, Modou Jobe, Amelia C. Crampin, Andrew M. Prentice, Janet Seeley, Edith F. Chikumbu, Joseph Mugisha, Ronald Makanga, Albert Dube, Frances S. Mair, Bhautesh Dinesh Jani.

**Writing – original draft:** Alison J. Price, Bhautesh Dinesh Jani.

**Writing – review & editing:** Alison J. Price, Modou Jobe, Isaac Sekitoleko, Amelia C. Crampin, Andrew M. Prentice, Janet Seeley, Edith F. Chikumbu, Joseph Mugisha, Ronald Makanga, Albert Dube, Frances S. Mair, Bhautesh Dinesh Jani.

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
