## [Decision Letter · Decision Letter 0]

4 Jul 2023

PGPH-D-23-00822

Epidemiology of Multimorbidity in low-income countries of sub-Saharan Africa: findings from four population cohorts

Dear Dr.Jani,

Thank you for submitting your manuscript to PLOS Global Public Health. After careful consideration, we feel that it has merit but does not fully meet PLOS Global Public Health’s publication criteria as it currently stands. Therefore, we invite you to submit a revised version of the manuscript that addresses the points raised during the review process.

EDITOR: Please explain in detail the accessibility of the data used for this study as per PLOS Global Public Health data availability policy. What is currently written does not meet the data availability policy. Please follow the link Data Availability | PLOS Global Public Health for guidance as to what is required. 

Please submit your revised manuscript by 17th July 2023. If you will need more time than this to complete your revisions, please reply to this message or contact the journal office at globalpubhealth@plos.org. Please include the following items when submitting your revised manuscript:

We look forward to receiving your revised manuscript.

Kind regards,

Peter Bai James, PhD

Academic Editor

Journal Requirements:

2. Please send a completed 'Competing Interests' statement, including any COIs declared by your co-authors. If you have no competing interests to declare, please state "The authors have declared that no competing interests exist". Otherwise please declare all competing interests beginning with twhe statement "I have read the journal's policy and the authors of this manuscript have the following competing interests:"

3. Please provide separate figure files in .tif or .eps format only and remove any figures embedded in your manuscript file. Please also ensure all files are under our size limit of 10MB.

Additional Editor Comments (if provided):

Reviewers' comments:

Reviewer's Responses to Questions

**Comments to the Author**

1. Does this manuscript meet PLOS Global Public Health’s publication criteria? Is the manuscript technically sound, and do the data support the conclusions? The manuscript must describe methodologically and ethically rigorous research with conclusions that are appropriately drawn based on the data presented.

Reviewer #1: Yes

Reviewer #2: Yes

Reviewer #3: Yes

2. Has the statistical analysis been performed appropriately and rigorously?

Reviewer #1: Yes

Reviewer #2: Yes

Reviewer #3: Yes

3. Have the authors made all data underlying the findings in their manuscript fully available (please refer to the Data Availability Statement at the start of the manuscript PDF file)?

Reviewer #1: Yes

Reviewer #2: No

Reviewer #3: No

4. Is the manuscript presented in an intelligible fashion and written in standard English?

Reviewer #1: No

Reviewer #2: Yes

Reviewer #3: Yes

5. Review Comments to the Author

Reviewer #1: The authors describe the epidemiology of multimorbidity in three low income countries in sub-Saharan Africa. The manuscript is generally well written, but I have some minor suggestions to assist in improving readability and clarity.

1. There are several typographical errors including stray upper case letters, inappropriate punctuation and a repeated phrase on page 9 (definition of asthma and epilepsy) that need to be addressed.

2. Please clarify whether the blood samples taken from participants in The Gambia were fasting samples and if they were analysed for this study or if results were extracted from the database.

3. Please clarify whether blood samples were taken in the Uganda cohort and if so, how they were handled and analysed.

4. Similarly, please clarify how all blood samples whose results are included in the definition of multimorbidity were analysed.

5. Please clarify why a previous diagnosis of hypertension was not used as a criterion for defining hypertension.

6. Please clarify whether HIV was included in the definition of multimorbidity. It is not included in the list on page 9, but is included subsequently.

7. Please justify why pregnant women were only excluded from the anthropometric analyses and not from the study as a whole or a sensitivity analysis performed excluding pregnant women. Pregnancy can affect parameters such as blood glucose and blood pressure.

8. The authors use the phrase "materially greater" several times in the manuscript-please clarify how this was determined.

9. The authors assert that females were at higher risk of multimorbidity in all study populations-this was not however the case in The Gambia.

10. Citation 40 is out of sequence in the discussion.

11. Please ensure that use of upper case letters is standardised in the reference titles.

12. I suggest citing numbers to 1 decimal place in text and tables.

13. i suggest switching around the row and column variables in Tables 1 and 2.

14. Please use the term "High cholesterol" in Table 4, rather than just "Cholesterol".

15. In Table 5, I suggest explaining that estimates are incidence rate ratios in a table footnote rather than including this in a row.

16. In Table 5, I suggest placing the p values in appropriately placed columns rather than rows.

Reviewer #2: A relevant and timely paper on the epidemiology of multimorbidity in sub-Saharan African countries that addresses research gaps.

Minor

1. Please add line numbers to your revised manuscript and refer to these when highlighting corrections in your response to reviewers

2. page 6 methods paragraph 1: add country name in the initial intro to the datasets (e.g. "Kiang West Longitudinal

Population Study (The Gambia)...")

3. page 12 Table 1: standard for sociodemographic indicators to be in left-hand column, and country data to be in subsequent columns (i.e. table rotated -90degrees) - same for other tables in document which are in this format. Column "female number" would better read as "female respondents"

4. page 15 table 3: combine rows with number of cases and crude prevalence into one (XX (XX%)) for easier reading

Moderate

5. We tend to overestimate the impact of smoking and alcohol consumption on morbidity in non-western contexts as this relationship is well established in caucasian populations in high income countries. However there are very different genetic and cultural drivers of morbidity and I would have liked to see this better highlighted: for instance whilst the authors do mention level of active lifestyles and diet in their discussion as possible additional risk factors, it is not clear whether such information was collected and whether it could be used to better inform the models in table 5. This is underlined by the data from Malawi in particular - the lowest levels of alcohol and smoking of the three included countries, but had a trend of the highest prevalences of morbidity. The authors are probably already aware of this introductory synopsis to the Hebe Gouda study (and the Gouda study itself):

https://www.thelancet.com/journals/langlo/article/PIIS2214-109X(19)30370-5/fulltext

which also supports the relevance of risk factors such as diet, air pollution, and exercise over alcohol and smoking.

If no such data is available, It would be great to see an expanded discussion of the likely additional risk factors that are SSA or LMIC specific.

Reviewer #3: Thank you for the opportunity to review the manuscript entitled, “Epidemiology of Multimorbidity in low-income countries of sub-Saharan Africa: findings from four population cohorts.” The authors present interesting data around common morbidities across a series of cohorts in 4 countries and provide age-standardized prevalence; the graphic (figure 1) is very useful and clear. Further, they present a series of potential risk and demographic factors. Interestingly, some of the expected relationships were inverse in some populations (e.g. higher education associated with higher prevalence of morbidity). There are a few minor areas for consideration and clarification.

MAJOR:

• In alignment with PLoS Global Public Health journal requirements, the authors need to clarify the availability of data. The current statement says, “The data that support the findings of this study are available from respective data controllers subject to successful registration and data governance application process.” The authors need to specify where data can be found and who controls the access process. This should be added in the data availability statement with weblinks/webpages or the process for requesting data.

• In Table 4, the authors calculate the multimorbidity prevalence (any combination of 2 or more LTC) by country. However, each country had a different number of underlying morbidities assessed. As part of the limitations, can the authors provide any potential risk or limitation of comparing these prevalence estimates across countries? In Malawi there are more morbidities assessed and therefore, higher likelihood for these participants to be classified as having met the multimorbidity definition. If these authors do not think this is a risk or potential source of bias, please provide further detail.

MINOR:

• In the Introduction, pleases clarify which “Academy of Medical Sciences” is referenced.

• Please consider adding the country names in the first paragraph of the ”Materials and Methods” section for consistency and specificity.

• It is a bold statement to say “this is the first three country comparative study” when there are several systematic reviews noted and data from other African countries highlighted in the discussion. Consider rephrasing or be clear about where searches were conducted to state that this is the first study.

• In the strengths and limitations section, it would be useful it the authors noted directionality of the impact on estimates by the differences in protocol for measurement of biomarker and anthropometric assessment.

6. PLOS authors have the option to publish the peer review history of their article (what does this mean?). If published, this will include your full peer review and any attached files.

**Do you want your identity to be public for this peer review?** For information about this choice, including consent withdrawal, please see our Privacy Policy.

Reviewer #1: No

Reviewer #2: No

Reviewer #3: No

---

## [Decision Letter · Decision Letter 1]

8 Nov 2023

Epidemiology of Multimorbidity in low-income countries of sub-Saharan Africa: findings from four population cohorts

PGPH-D-23-00822R1

Dear Dr Bhautesh Dinesh Jani,

We are pleased to inform you that your manuscript 'Epidemiology of Multimorbidity in low-income countries of sub-Saharan Africa: findings from four population cohorts' has been provisionally accepted for publication in PLOS Global Public Health.

Best regards,

Peter Bai James, PhD

Academic Editor

Reviewer Comments (if any, and for reference):

Reviewer's Responses to Questions

**Comments to the Author**

1. If the authors have adequately addressed your comments raised in a previous round of review and you feel that this manuscript is now acceptable for publication, you may indicate that here to bypass the “Comments to the Author” section, enter your conflict of interest statement in the “Confidential to Editor” section, and submit your "Accept" recommendation.

Reviewer #1: All comments have been addressed

Reviewer #3: All comments have been addressed

2. Does this manuscript meet PLOS Global Public Health’s publication criteria? Is the manuscript technically sound, and do the data support the conclusions? The manuscript must describe methodologically and ethically rigorous research with conclusions that are appropriately drawn based on the data presented.

Reviewer #1: Yes

Reviewer #3: Yes

3. Has the statistical analysis been performed appropriately and rigorously?

Reviewer #1: Yes

Reviewer #3: Yes

4. Have the authors made all data underlying the findings in their manuscript fully available (please refer to the Data Availability Statement at the start of the manuscript PDF file)?

Reviewer #1: Yes

Reviewer #3: Yes

5. Is the manuscript presented in an intelligible fashion and written in standard English?

Reviewer #1: Yes

Reviewer #3: Yes

6. Review Comments to the Author

Reviewer #1: All of my comments have been addressed.

Reviewer #3: Thank you for thoroughly addressing the reviewers’ comments!

7. PLOS authors have the option to publish the peer review history of their article (what does this mean?). If published, this will include your full peer review and any attached files.

**Do you want your identity to be public for this peer review?** For information about this choice, including consent withdrawal, please see our Privacy Policy.

Reviewer #1: No

Reviewer #3: No
